# Total Variation Weighted Low-Rank Constraint for Infrared Dim Small Target Detection

**Xiaolong Chen** [1,2,3], **Wei Xu** [1,3,*], **Shuping Tao** [1,3], **Tan Gao** [1,2,3], **Qinping Feng** [1,3] and **Yongjie Piao** [1,3]

1   Changchun Institute of Optics, Fine Mechanics and Physics, Chinese Academy of Sciences, Changchun 130033, China
2   University of Chinese Academy of Sciences, Beijing 100039, China
3   Key Laboratory of Space-Based Dynamics and Rapid Optical Imaging Technology, Chinese Academy of Sciences, Changchun 130033, China
*   Correspondence: xwciomp@126.com; Tel.: +86-0431-85686367

**Abstract:** Infrared dim small target detection is the critical technology in the situational awareness field currently. The detection algorithm of the infrared patch image (IPI) model combined with the total variation term is a recent research hotspot in this field, but there is an obvious staircase effect in target detection, which reduces the detection accuracy to some extent. This paper further investigates the problem of accurate detection of infrared dim small targets and a novel method based on total variation weighted low-rank constraint (TVWLR) is proposed. According to the overlapping edge information of image background structure characteristics, the weights of constraint low-rank items are adaptively determined to effectively suppress the staircase effect and enhance the details. Moreover, an optimization algorithm combined with the augmented Lagrange multiplier method is proposed to solve the established TVWLR model. Finally, the experimental results of multiple sequence images indicate that the proposed algorithm has obvious improvements in detection accuracy, including receiver operating characteristic (ROC) curve, background suppression factor (BSF) and signal-to-clutter ratio gain (SCRG). Furthermore, the proposed method has stronger robustness under complex background conditions such as buildings and trees.

**Keywords:** overlapping edge information; infrared small target detection; low-rank constraint; total variational regularization



## 1. Introduction

The infrared imaging system uses the target radiation received by the sensor to image, which has the advantages of being unaffected by the environment, small size and passive imaging. As a crucial technique of situation awareness, the detection and tracking of infrared dim and small targets play a significant role in the precision strike, perception and early warning systems [1]; this has been extensively studied [2–4]. Generally, the overall contrast and signal-to-noise ratio of infrared images are low and the target is easily submerged in the clutter background and noise interference, which is difficult to distinguish. Furthermore, the number of pixels occupied by the target is small. The Society of Photo-optical Instrumentation Engineers (SPIE) defines that the imaging area of infrared dim and small target is less than 0.12% of the total pixel number, no more than 81 pixels on an image of 256 × 256 [5]. However, in the actual imaging process, the target is even much smaller than this value and lacks shape and texture features, which results in a significant increase in the difficulty of target detection. Therefore, infrared dim target-detection technology has become a challenging and hot topic.

Generally, infrared dim small target-detection algorithms are mainly divided into multi-frame detection and single-frame detection. Multi-frame detection uses the continuity and correlation of moving targets in multi-frame images to achieve detection, while single-frame detection mainly uses the single-frame image to extract the gradient, grayscale,

contrast and other characteristics of the small target. Compared with multi-frame detection, it has the advantages of low complexity, high execution efficiency and easy hardware implementation.

The traditional single-frame-detection algorithm divides an infrared image into the background region, the target region and the noise region and the model is as follows:

$$f_D(i,j) = f_E(i,j) + f_T(i,j) + f_B(i,j) \tag{1}$$

where $(i,j)$, $f_D$, $f_E$, $f_T$ and $f_B$ represent the position of a specific pixel, the infrared small target image, noise region, target region and background region, respectively.

It can be seen that the performance of conventional algorithms depends on assumptions about the background and target, which has great limitations. Since the image has nonlocal autocorrelation properties, $f_B(i,j)$ can be considered a low-rank matrix. In the meantime, since the target possesses very few image pixels, $f_T(i,j)$ can be treated as a sparse matrix. Therefore, the traditional infrared image model is extended to the infrared patch image (IPI) model.

The IPI model regards target detection as the optimization problem of separating sparse matrix and low-rank matrix and is solved by using principal component analysis. The following is the target image redefined by the IPI model:

$$D = E + T + B \tag{2}$$

where $D$, $E$, $T$ and $B$, respectively, represent the infrared patch image, noise region, target region and background region.

Neglecting the noise part, target detection is carried out by constraining the sparse matrix and the low-rank matrix, respectively:

$$\min_{B,T} \|B\|_* + \lambda \|T\|_1, \ \ s.t. D = T + B \tag{3}$$

where $\lambda$ is the positive balance parameter, $\|\cdot\|_*$ represents the nuclear norm and $\|\cdot\|_1$ represents the $l_1$ norm.

However, the sparse term is constrained by the $l_1$ norm that makes part of the background remain in the target image or excessively narrow the target. Some low brightness non-target points also show sparsity under the $l_1$ norm constraint, which causes false detection. Moreover, when there is a strong edge in the image, the target image will leave the edge residual and make the estimated background fuzzy. This paper proposes the TVWLR mode to address these issues. The overlapping edge information and total variation ($TV$) regularization term are combined to characterize the background structural features and the constraint of the low-rank term is strengthened to reduce the false detection rate of target detection. Meanwhile, we employ the adaptive weight constraint low-rank term to accurately evaluate the background image.

The following are the main contributions of this paper:

(1) Considering the problem that it is difficult to accurately detect targets in complicated backgrounds, a total variational weighted low-rank constraint method is proposed. The proposed method strengthens the constraints on low-rank terms, which can better evaluate the background image and improves target-detection probability.

(2) By applying overlapping edge information (OEI) to determine the weights that constrain low-rank terms, the staircase effect is effectively suppressed. Meanwhile, the $l_{2,1}$ norm is introduced to remove strong edges so as to solve the problem of false detection caused by low-brightness non-target points.

(3) An optimization algorithm combined with the alternating direction method of multipliers (ADMM) is given to resolve the TVWLR model accurately. Moreover, the solution process is simplified by using tolerance error as a stopping condition.

(4) We conduct many experiments on some of the scene images after determining the specific values of the pivotal parameters. The feasibility of the suggested method is verified by qualitative and quantitative analysis of the experimental results.

The following shows the organization of the remaining parts of this paper: Section 2 briefly introduces the related works on infrared small target detection; we describe in detail the process of proposing the TVWLR model and related optimization methods in Section 3; we carry out experiments on six sequential images and conduct qualitative and quantitative analysis, respectively, in Section 4; Section 5 is the discussion; Section 6 summarizes the conclusions of this paper.

## 2. Related Works

### 2.1. Sequence Image-Detection Methods

The methods require multi-frame image information, which leads to low detection efficiency and poor practicability. In the case of uniform background distribution, the methods such as dynamic programming [6], spatial filtering [7] and matched filtering [8] have good background suppression ability. However, the relative speed of the image detector and the target is fast in the actual application process, which makes it difficult to ensure that the image has a uniform background, resulting in poor detection performance [9].

### 2.2. Single-Frame Image-Detection Methods

The methods utilize gray and contrasting characteristics of the image to achieve detection; this involves low complexity and high detection efficiency. The methods are mainly composed of traditional filtering methods, methods based on human vision, optimization-based methods and method-based deep learning.

Traditional filtering methods such as Tophat transform [10], maximum mean and maximum median [11] utilize the residual image of the original image and the filtered image to achieve target enhancement and background noise suppression. Due to the background complexity of the actual application environment, the algorithms mentioned above usually cannot meet the detection accuracy requirements.

The methods based on human vision take the saliency of the target in the adjacent area as the detection basis. Based on the spatially discontinuous features of the target [12], Chen et al. developed the local contrast map (LCM) algorithm [13]. The gray difference of a 3 × 3 neighborhood is utilized to estimate the saliency of pixels in the neighborhood. To improve the detection speed of the algorithm, an improved local contrast metric method (ILCM) was proposed by Han et al. [14]. Based on the characteristics of bright and dark targets, Wei et al. created a multi-scale patch-based contrast method (MPCM) [15]. By using the matching filter and the principle of closest mean, Han et al. designed an enhanced closest-mean background estimation (ECMBE) model to suppress high brightness backgrounds and improve the signal-to-noise ratio [16]. Bai et al. fused the contrast measurement mapping derived from different derivative sub-bands and proposed a contrast measurement method based on derivative entropy (DECM) [17]. However, these methods depend on the brightness difference between the background and target and cannot achieve ideal detection results when the brightness difference is low.

The optimization-based methods treat target detection as an ill-posed inverse problem. Combining this idea, Gao et al. [18] created the IPI model, which exploited the nonlocal autocorrelation of the background to turn target detection into an optimization problem of the background matrix and the target matrix. According to the thermal characteristics of the target, Dai et al. developed a non-negative infrared patch-image model (NIPPS) [19,20]. Zhang et al. [21] introduced an advanced local prior graph that simultaneously encodes background-related and target-related information and proposed a detection method incorporating the partial sum of tensor kernel norm (PSTNN), which can significantly reduce the algorithm complexity and computational time. Wang et al. [22] designed the total variation regularization and principal component pursuit (TV-PCP) model to effectively preserve the background edge information. Zhang et al. [23] used self-regularization terms

to describe background features and devised the self-regularized weighted sparse model (SRWS). The above methods reach good detection results, but the detection accuracy of images with special strong edges is poor and the false alarm rate and missed detection rate are high.

The methods based on deep learning are the latest technology in the field of target detection. Wang et al. [24] adopted a dictionary learning method and considered the non-local characteristics of background and target and developed a more flexible stable multi-subspace learning model (SMSL). Shi et al. [25] designed a denoising autoencoder model (CDAE), which regarded small targets as noise, used a denoising autoencoder for denoising reconstruction and obtained a detection image by subtracting the original image from the reconstructed image. In order to improve the performance of network-detection targets, Du et al. [26] proposed a target-oriented shallow-deep features (TSDFs) model based on deep semantic features and shallow detail features of targets. Gao et al. [27] devised a feature mapping deep neural network (FMDNN) to solve the problem that small target features are difficult to extract. For star maps with non-uniform backgrounds, Xue et al. [28] designed a StarNet that employed pixel-level classification to quickly separate backgrounds and targets. To extract targets in cluttered backgrounds, Zhou et al. [29] proposed a 3D-based convolutional network that could reconstruct small targets. These methods show good detection ability. However, there are few infrared dim small target datasets publicly available at present, resulting in unsatisfactory robustness in diverse backgrounds.

## 3. Proposed Method

First, we briefly introduce the total variational models to characterize background features and preserve background information in this section. Second, we explain the concept and structure of overlapping edge information, which is utilized to constrain the image background and eliminate the staircase effect created by total variation. Third, the total variational weighted low-rank model and associated optimization algorithm are proposed. Finally, the quantitative evaluation metrics and qualitative evaluation methods are described.

### 3.1. TV Model

Rudin et al. [30] first presented a total variation model to remove image noise. This model smoothes the image inside the image and reduces the difference between adjacent pixels and as far as possible does not smooth the edge of the image. Therefore, it is an anisotropic model. If the infrared small target image is represented by $X \in R^{m \times n}$ and the pixel in row $i$ and column $j$ of image $X$ is $x_{i,j}$, the definition of $TV$ norm can be described by:

$$TV(X) = \sum_{j=1}^{n-1} |x_{m,j+1} - x_{m,j}| + \sum_{i=1}^{m-1} |x_{i+1,n} - x_{i,n}| + \sum_{j=1}^{n-1} \sum_{i=1}^{m-1} \sqrt{(x_{i,j+1} - x_{i,j})^2 + (x_{i+1,j} - x_{i,j})^2} \qquad (4)$$

It can be seen from Equation (4) that if the edge information is not considered, the total variation norm can be regarded as the $l_2$ norm of the image derivative. If we convert image $X$ to a column vector and use $P_i$ to represent the corresponding gradient operator, the discrete gradient of pixels at $i$ can be represented by $P_i \in R^2$. Therefore, the following non-differentiable, non-convex function can be obtained:

$$TV(X) = \sum_i \|P_i X\|_2 \qquad (5)$$

The total variational model is an effective regular item to maintain the image smoothness [31]. The model can reduce the disparity of the image to closely match the original image, remove unwanted details and retain crucial details such as edges. In addition, the total variation model can also accurately evaluate discontinuities in infrared images. Thus, we introduce total variation to characterize the image background features.

The total variation term enables detection algorithms to better preserve background information such as strong edges, which better estimate the background image. Some

sparse parts of non-target points are removed to reduce the false detection rate of target detection. However, the total variation model will appear as a significant staircase effect in practical applications [32–34], which makes it difficult to accurately detect the target.

### 3.2. Overlapping Edge Information

To address the staircase effect problem, the structural features of the image are characterized by OEI. It can be found from Figure 1 that the edge portion of the OEI feature image is rather visible and numerous. Based on this property, we utilize OEI to obtain the equivalent weight to constrain low-rank terms that suppress the staircase effect. To get the OEI of image $X$, the matrix $Q$ is obtained by combining the overlapping matrix of horizontal and vertical derivatives of the image:

$$Q(i,j) = |Q_v(i,j)| + |Q_h(i,j)| \tag{6}$$

where $Q_v(i,j) = \sum_{i=-m_1}^{m_2} \sum_{j=-m_1}^{m_2} G_v(i,j)$, $Q_h(i,j) = \sum_{i=-m_1}^{m_2} \sum_{j=-m_1}^{m_2} G_h(i,j)$, $G_v(i,j)$ and $G_h(i,j)$ represent the first derivative of pixel $(i,j)$ in the vertical direction and horizontal direction, respectively. Among them, $m_1 = \left[\frac{l-1}{2}\right]$, $m_2 = \left[\frac{l}{2}\right]$ and $l$ represents the number of overlapping information groups; operator $[n]$ is the largest integer equal to or less than the number $n$.

The smaller the element difference in the OEI, the better it can characterize the structural features and suppress the background. Therefore, we use OEI to constrain the low-rank term to highlight the target in the image, thereby improving the background suppression ability and detection accuracy of the detection algorithm.

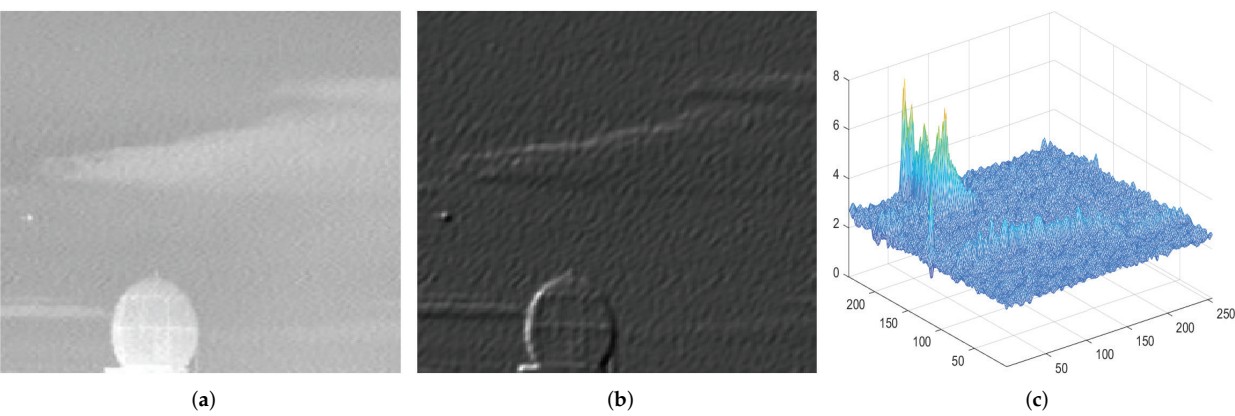

(a)      (b)      (c)

**Figure 1.** The featured image of OEI. (**a**) original image. (**b**) OEI feature image. (**c**) three-dimensional image of OEI.

### 3.3. TVWLR Model

As shown in Figure 2, there are large residuals in the target image detected by the IPI model, which can cause false detection. To improve this phenomenon, we introduce the total variation regularization term into the IPI model to retain the background edge well:

$$\min_{B,T} \|B\|_* + \lambda_1 TV(B) + \lambda_2 \|T\|_1,$$
$$s.t. D = E + T + B, \ \|E\|_F \leq \delta \tag{7}$$

where $\|\cdot\|_F$ represents the $F$ norm, $TV(\cdot)$ represents the $TV$ norm, $\lambda_1$ and $\lambda_2$ are two positive balance parameters, $\delta$ is a positive parameter that changes with the image.

According to Section 3.1, the total variational term is represented by Equation (5):

$$\min_{B,T} \|B\|_* + \lambda_1 \sum_i \|P_i B\|_2 + \lambda_2 \|T\|_1,$$
$$s.t.D = E + T + B, \|E\|_F \leq \delta \tag{8}$$

where $P_i$ represents the gradient operator. Both the nuclear norm and the *TV* norm constrain the background, which can obtain better background estimation.

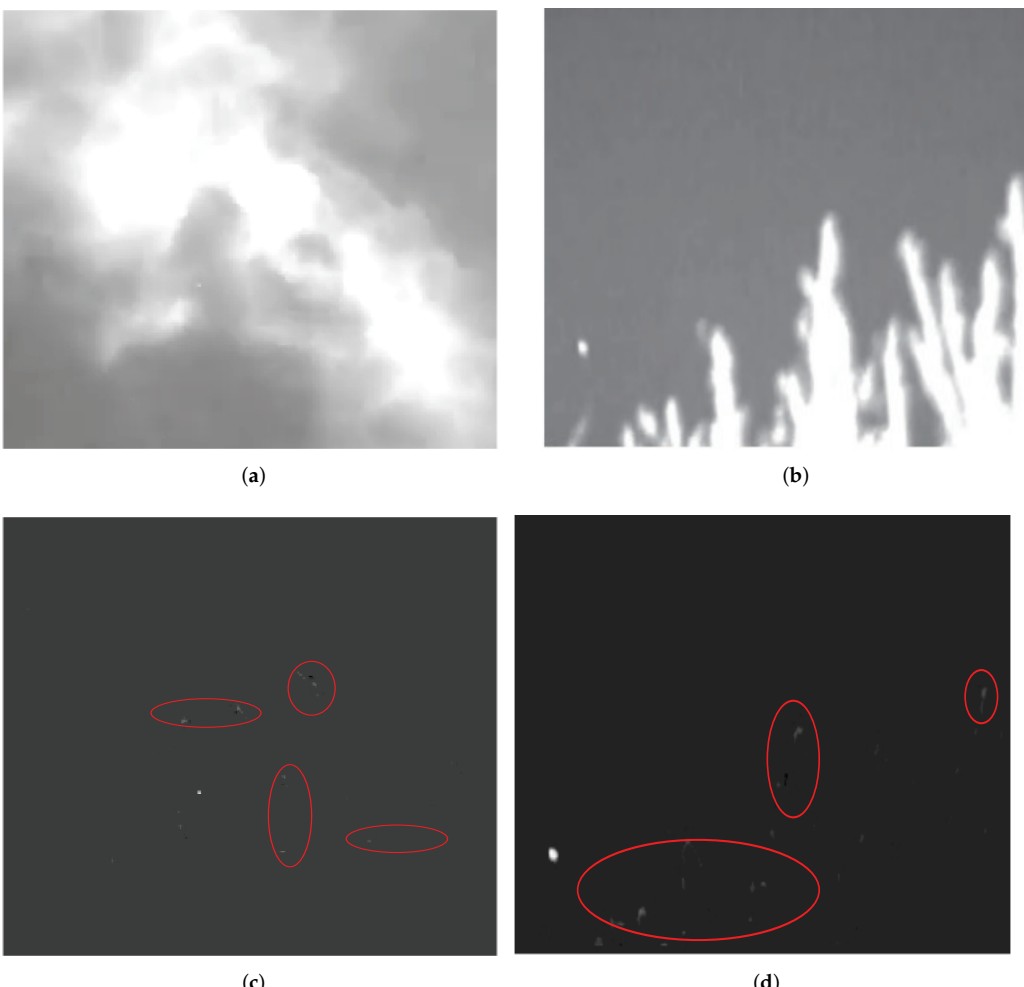

**Figure 2.** The IPI model detection results. (**a**,**b**) original images. (**c**,**d**) target images with clutter.

According to the background structure information, the low-rank term is weighted to suppress the staircase effect so as to accurately characterize the image background feature and obtain a clear background estimation. First, the matrix $Q$ is converted into patch image $Q$. Then, the following weight equation is obtained by the OEI of the infrared image:

$$\omega = \exp(\alpha * \frac{Q - Q_{\min}}{Q_{\max} - Q_{\min}}) \tag{9}$$

where $\alpha$ is the tensile coefficient, $Q_{\min}$ and $Q_{\max}$ the minimum value and the maximum value of the matrix $Q$, respectively.

In order to remove the residuals left by the strong edges of a complex infrared image in the target image, we introduce the following defined $l_{2,1}$ norm:

$$\|E\|_{2,1} = \sum_i \sqrt{\sum_j E_{ij}^2} \tag{10}$$

In summary, the proposed total variation weighted low-rank constraint (TVWLR) model is as follows:

$$\min \|B\|_{\omega,*} + \lambda_1 \sum_i \|P_i B\|_2 + \lambda_2 \|T\|_1 + \beta \|E\|_{2,1},$$
$$s.t.D = E + T + B \tag{11}$$

where $P_i$ represents the gradient operator; $\beta$ is the penalty factor. We obtain the detected target image and related target information after solving Equation (11).

Figure 3 describes the framework of the proposed method:

1. Specify a sliding window and step size, obtain each patch in turn and then vectorize these patches into the column vectors to form a new matrix, thereby obtaining the patch image.

2. Calculate the OEI of the original image and then use the same step and sliding window size as the previous step to obtain the patch weight.

3. Initialize the relevant parameters, input the patch image and patch weight into Algorithm 1 and solve it through the designed optimization algorithm.

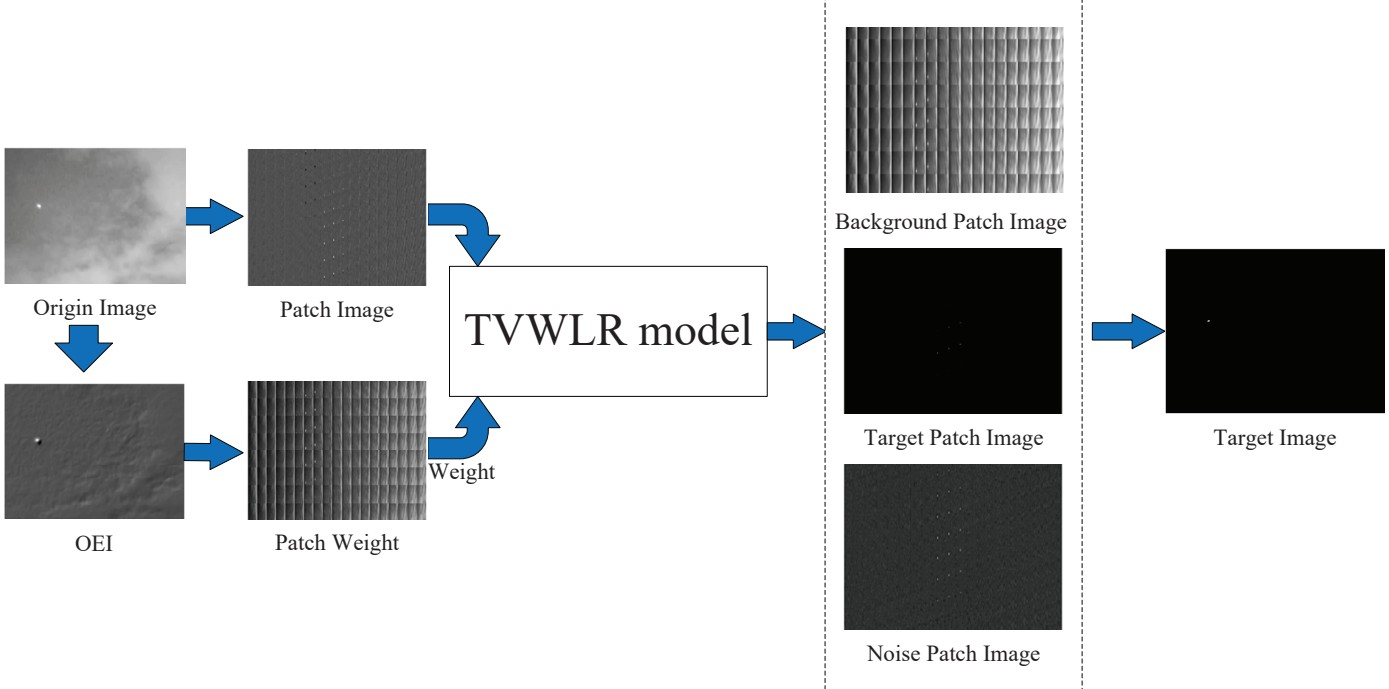

**Figure 3.** The framework of the proposed method.

---

**Algorithm 1:** The solution process of the TVWLR model.

---

**Input:** Original patch image $D$, $\lambda_1$, $\lambda_2$, $\beta$, $\mu^0$; these parameters are used as inputs
    and the experimental part mentioned how to obtain the values of $\lambda_1$, $\lambda_2$, $\beta$, $\mu$;

**Output:** $B$, $T$;

**Initialize:** $Z_1^0 = B^0 = T^0 = E^0 = zeros(m, n)$, $\rho = 1.1$, $t = 0$,
  $Y_1 = Y_3 = zeros(m, n)$, $Z_2^0 = zeros(mn, 2)$, $Y_2 = zeros(2, mn)$;

**while** *not converged* **do**

  $Z_1^{t+1}$ is solved by singular value threshold Equation (15);

  $Z_2^{t+1}$ is updated by the two-dimensional Shrinkage-like Equation (17);

  $B^{t+1}$ is updated through Equation (19);

  $T^{t+1} = \ell_{\frac{\lambda_2}{\mu^t}}\left(D - B^{t+1} - E^t - \frac{Y_3^t}{\mu^t}\right)$;

  $E^{t+1}(:, i) = \begin{cases} \dfrac{\|M(:,i)\|_2 - \frac{\beta}{\mu^t}}{\|M(:,i)\|_2} M(:,i) & if \|M(:,i)\|_2 > \frac{\beta}{\mu^t} \\ 0 & otherwise \end{cases}$ ;

  Update others: $\begin{array}{l} Y_1^{t+1} = Y_1^t + \mu^t(Z_1^{t+1} - B^{t+1}), \\ Y_2^{t+1} = Y_2^t + \mu^t(Z_2^{t+1} - DB^{t+1}), \\ Y_3^{t+1} = Y_3^t + \mu^t(D - B^{t+1} - T^{t+1} - E^{t+1}) \end{array}$ ;

  $\mu^{t+1} = \rho\mu^t$;

  Update $t$: $t = t + 1$;

**end**

---

*3.4. Optimization Algorithm*

We propose an optimization method by combining ADMM to solve Equation (11) in this section. First, Equation (11) is equivalent to:

$$\begin{aligned} \min \; & \|Z_1\|_{\omega,*} + \lambda_1 \sum_i \|z_i\|_2 + \lambda_2 \|T\|_1 + \beta\|E\|_{2,1}, \\ s.t. \; & Z_1 = B \\ & Z_2 = [z_1, z_2, \ldots, z_{mn}], \; z_i = P_i B \\ & D = B + T + E \end{aligned} \tag{12}$$

Then, Equation (12) is transformed into the augmented Lagrange function:

$$\begin{aligned} L_A = \; & \|Z_1\|_{\omega,*} + \lambda_1 \sum_i \|z_i\|_2 + \lambda_2 \|T\|_1 + \beta\|E\|_{2,1} + \langle Y_1, Z_1 - B \rangle \\ & + \frac{\mu}{2}\|Z_1 - B\|_F^2 + \sum_i \left(\langle y_i, z_i - P_i B \rangle + \frac{\mu_i}{2}\|z_i - P_i B\|_F^2\right) \\ & + \langle Y_3, D - T - B - E \rangle + \frac{\mu}{2}\|D - T - B - E\|_F^2 \end{aligned} \tag{13}$$

where $\|\cdot\|_F$ is the $F$ norm, $\langle \cdot, \cdot \rangle$ represents the interior product of two matrices, $Y_1$, $Y_3$ and $Y_2(Y_2 = [y_1, y_2, \ldots, y_{mn}] \in R^{2 \times mn})$ denote the Lagrange multiplier, $\mu$ is the penalty factor.

Equation (13) can be solved iteratively with our designed optimization algorithm. When the $(t + 1)th$ iteration is performed:

$$Z_1^{t+1} = \arg\min_{Z_1} \|Z_1\|_{\omega,*} + \langle Y_1, Z_1 - B \rangle + \frac{\mu^t}{2}\left\|Z_1 - B^k\right\|_F^2 \tag{14}$$

The singular value threshold method can be used to solve Equation (14). The following is the singular threshold function:

$$\begin{aligned} SVT_\varepsilon(M) = \; & Udiag[(\tau - \varepsilon)_+]V^T, \\ (\tau - \varepsilon)_+ = \; & \begin{cases} \tau - \varepsilon & \tau > \varepsilon \\ 0 & otherwise \end{cases} \end{aligned} \tag{15}$$

where $M = U \sum V^T$ represents the singular value decomposition of matrix $M$.

$$Z_2^{t+1} = \arg\min_{Z_2} \sum_i \left( \|z_i\|_2 + \langle y_i, z_i - P_i B \rangle + \frac{\mu_i^t}{2} \|z_i - P_i B\|_F^2 \right) \tag{16}$$

According to Ref. [35], Equation (16) can be solved by using a two-dimensional shrinkage-like formula:

$$z_i = \max\left\{ \left\| P_i B - \frac{y_i}{\mu_i^t} \right\|_2 - \frac{1}{\mu_i^t}, 0 \right\} \cdot \frac{\left( P_i B - \frac{y_i}{\mu_i^t} \right)}{\left\| P_i B - \frac{y_i}{\mu_i^t} \right\|_2} \tag{17}$$

The solution process of $B^{t+1}$ is as follows:

$$B^{t+1} \leftarrow \frac{\partial L_A}{\partial B} = 0 \tag{18}$$

Equation (18) is a linear problem and its solution process is as follows:

$$B^{t+1} = \frac{[Y_1^t + Y_3^t + \sum_i \left( D_i^T y_i + \mu_i D_i^T z_i \right) + \mu \left( Z_1^{t+1} + D - T^{t+1} - E^{t+1} \right)]}{2\mu + \sum_i \mu_i D_i^T D_i} \tag{19}$$

Updates to $E^{t+1}$ and $T^{t+1}$ are as follows:

$$E^{t+1} = \arg\min_E \beta \|E\|_{2,1} + \frac{\mu^t}{2} \left\| D - B^{t+1} - T^{t+1} - E - \frac{Y_3^t}{\mu^t} \right\|_F^2 \tag{20}$$

$$T^{t+1} = \arg\min_T \lambda_2 \|T\|_1 + \frac{\mu^t}{2} \left\| D - B^{t+1} - T - E^t - \frac{Y_3^t}{\mu^t} \right\|_F^2 \tag{21}$$

According to Ref. [36] and Ref. [9], Equations (20) and (21) are solved, respectively:

$$E^{t+1}(:,i) = \begin{cases} \frac{\|M(:,i)\|_2 - \frac{\beta}{\mu^t}}{\|M(:,i)\|_2} M(:,i) & if \|M(:,i)\|_2 > \frac{\beta}{\mu^t} \\ 0 & otherwise \end{cases} \tag{22}$$

$$T^{t+1} = \ell_{\frac{\lambda_2}{\mu^t}} \left( D - B^{t+1} - E^t - \frac{Y_3^t}{\mu^t} \right) \tag{23}$$

In Equation (22), $M = D - B^{t+1} - T^{t+1} - \frac{Y_3^t}{\mu^t}$. In Equation (23), $\ell_\varepsilon(\cdot)$ represents the soft threshold operation [37].

Updates to $Y_i^{t+1}$ and $\mu^{t+1}$ are as follows:

$$\begin{aligned} Y_1^{t+1} &= Y_1^t + \mu^t (Z_1^{t+1} - B^{t+1}), \\ Y_2^{t+1} &= Y_2^t + \mu^t (Z_2^{t+1} - DB^{t+1}), \\ Y_3^{t+1} &= Y_3^t + \mu^t (D - B^{t+1} - T^{t+1} - E^{t+1}) \end{aligned} \tag{24}$$

$$\mu^{t+1} = \rho\mu^t \tag{25}$$

where $\rho > 0$.

Finally, we describe the whole iterative optimization process in Algorithm 1.

### 3.5. Evaluation Metrics

We introduce the definitions of several evaluation metrics in this subsection, including receiver operating characteristic (ROC) curve, background suppressor factor (BSF) and signal-to-clutter ratio gain (SCRG).

SCRG and BSF are good evaluations of the ability of detection target and background suppression and can be expressed by the following two formulas:

$$SCRG = \frac{S_{out}/C_{out}}{S_{in}/C_{in}} \tag{26}$$

$$BSF = \frac{C_{in}}{C_{out}} \tag{27}$$

where $C_{in}$ and $C_{out}$ represent the standard deviation of the background region of the original infrared image and the output target image, respectively. $S_{in}$ and $S_{out}$ represent the amplitude of the target region of the original infrared image and the detected target image, respectively.

$$S = T_{\max} - T_{\min} \tag{28}$$

where $T_{\max}$ and $T_{\min}$, respectively, are the maximum and minimum gray values of the target region.

In order to avoid infinity (Inf) [38] when calculating SCRG and BSF, as shown in Figure 4, we adopt the definition of background region in Ref. [23], as shown in Figure 4. The red square and black square in the figure represent the target region of size $a$ and the background region of size $d$, respectively. To ensure that all target pixels are included in the selected region, we set $a = 11$ and $d = 81$.

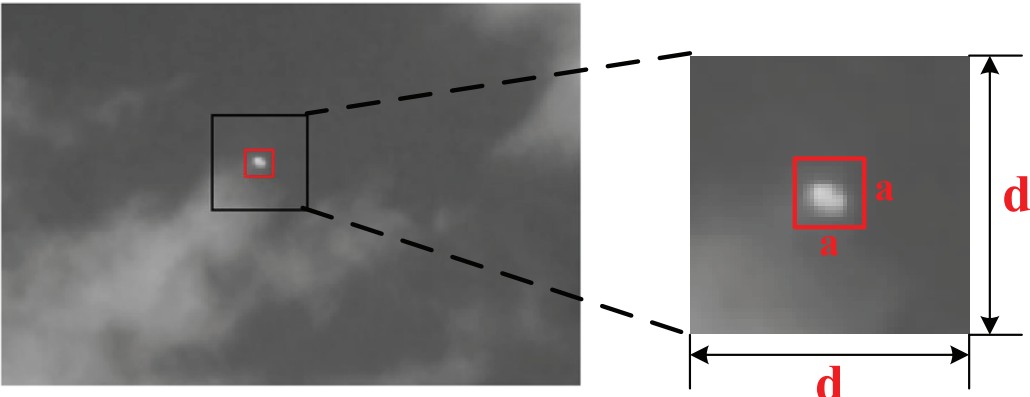

**Figure 4.** The background region and target region of the infrared small image.

To comprehensively assess the detection capability of all methods, two important evaluation matrices are introduced: the false alarm rate $F_a$ and the probability of detection $P_d$, which are expressed by the following two formulas:

$$F_a = \frac{N_f}{N_I} \tag{29}$$

$$P_d = \frac{N_t}{N_T} \tag{30}$$

where $N_f$ and $N_I$ represent the number of small targets detected incorrectly and the number of images, respectively; $N_t$ and $N_T$ represent the number of small targets actually detected and the actual number of small targets, respectively.

When drawing the ROC curve, we use $F_a$ and $P_d$ as the horizontal axis and vertical axis, respectively. Therefore, as the ROC curve approaches the upper left corner, the target detection ability improves. We also quantify the detection effect by calculating the area under the ROC curve (AUC) of all methods. Generally, the better the effect of target detection, the higher the AUC.

## 4. Experiments and Results

Firstly, the experimental parameters are determined, then we compare the TVWLR model with the other seven baseline methods.

### 4.1. Parameter Setting

The main parameters that influence the proposed method's detection performance are determined in this part. The low-rank term and the *TV* regularization term are both

balanced by $\lambda_1$. It is an empirical value of around 0.01, so we set it to 0.005. $\lambda_2$ is utilized to balance the mutual influence between the target region and background region. Considering the influence of the $TV$ term, we use the value $\lambda_2 = \dfrac{150}{\sqrt{\max(p,q)}}$ in the experiment, where $p$ and $q$, respectively, represent the width and length of the original patch image. $\mu$ directly affects the soft threshold operator of the calculation target and the convergence speed of the iterative process, which is a penalty operator. If $\mu$ is too small, the target will not be recognized; if $\mu$ is too large, the target image will have a lot of noise. To make $\mu$ change adaptively, we set $\mu = z\sqrt{\max(m,n)}$, where we set the range of $\mu$ from 0.5 to 5. For six sequential images, the detection performance is best when $z = 2$ or $z = 3$. In order to ensure the convergence speed, we choose $z = 3$. For the proposed method, we define the tolerance error as follows:

$$tol = \frac{\left\| D - T^t - B^t - E^t \right\|_F}{\|D\|_F} \tag{31}$$

where $t$ represents the number of the iterative process of the optimization method. The iterative process stops when $tol < 10^{-5}$, which is considered convergent.

*4.2. Experimental Preparation*

First, two scenes in Figure 2 are tested with the proposed method, which verifies the effectiveness of our detection algorithm. The processing results and the corresponding three-dimensional views are shown in Figure 5, where the red box marks the target.

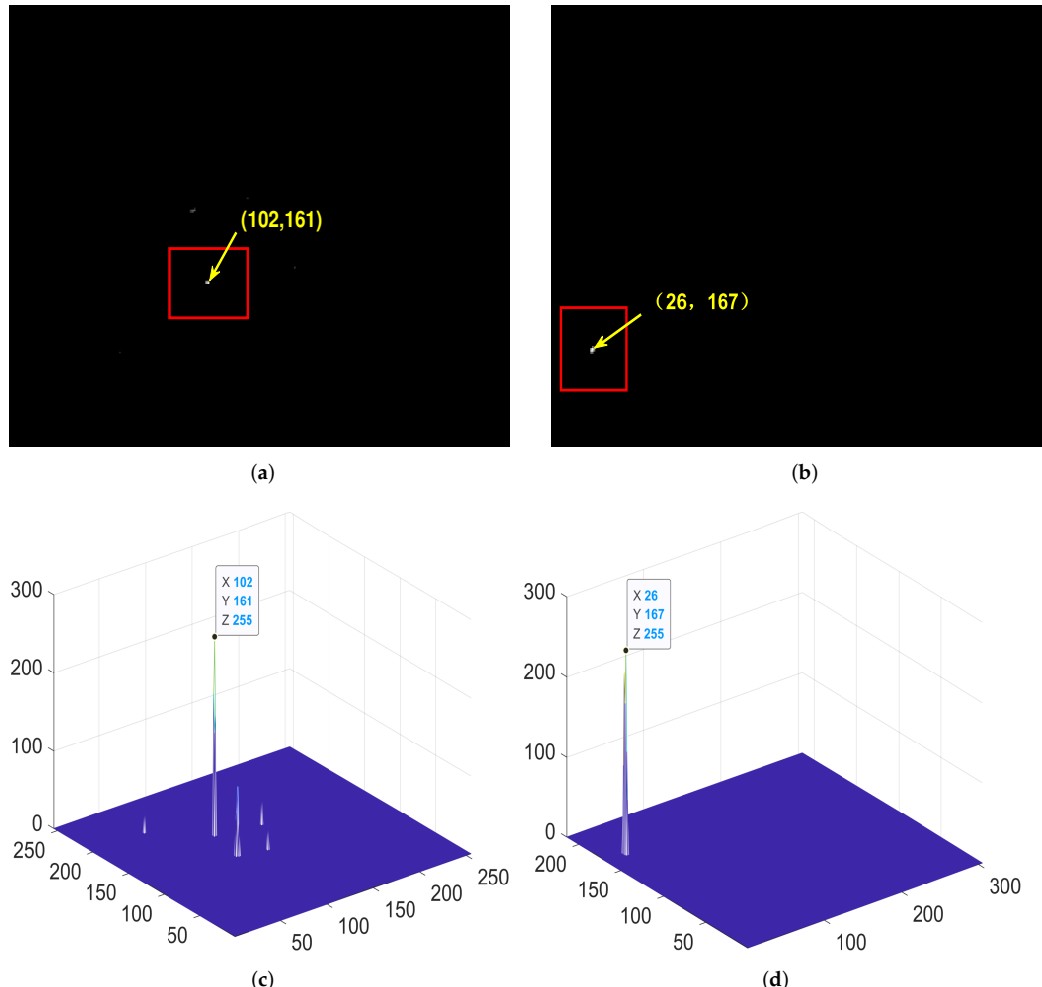

**Figure 5.** The TVWLR model detection results. (**a**,**b**) the target images processed by the TVWLR model. (**c**,**d**) the target images processed by the TVWLR model.

In order to further test the performance of the proposed model, a large number of experiments are performed in this paper. The experimental scenes include high-light clutter, cloud background, ground, sea level, etc., and the results of some of the scenes are selected for display. These six scenes and three-dimensional views are shown in Figures 6 and 7, respectively. These images feature various target sizes, diverse backgrounds and low signal-to-noise ratios, which make it difficult to successfully detect the target using traditional methods.

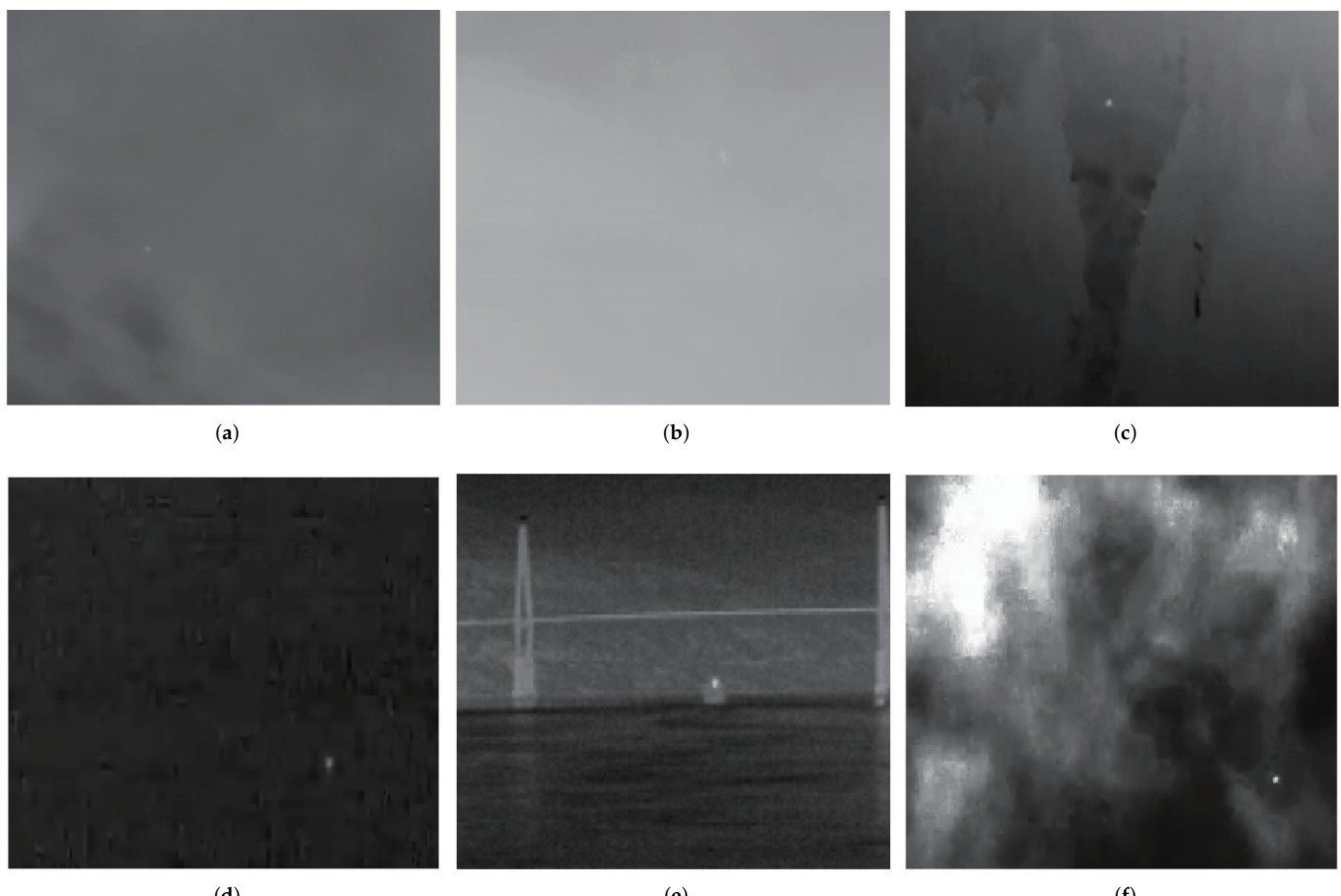

**Figure 6.** Infrared images of some real scenes. (**a**) Scene 1 (Seq1). (**b**) Scene 2 (Seq2). (**c**) Scene 3 (Seq3). (**d**) Scene 4 (Seq4). (**e**) Scene 5 (Seq5). (**f**) Scene 6 (Seq6).

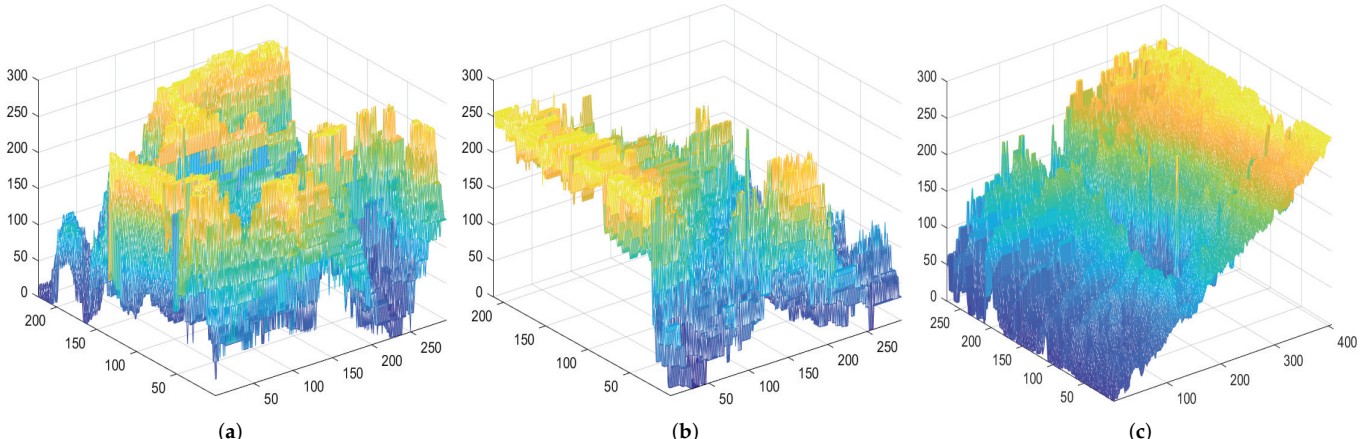

**Figure 7.** *Cont.*

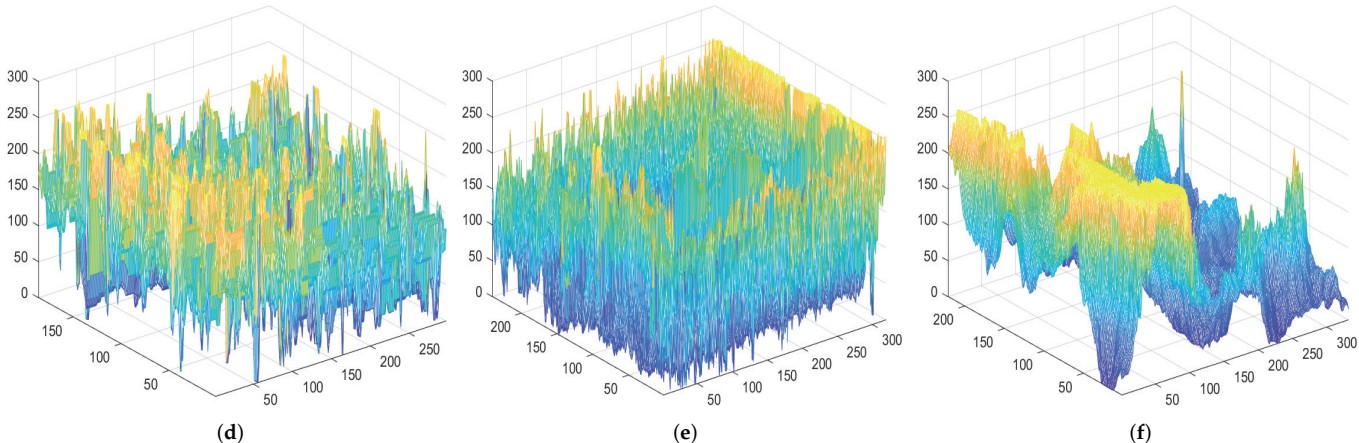

**Figure 7.** Infrared images of some real scenes. (**a**) Scene 1 (Seq1). (**b**) Scene 2 (Seq2). (**c**) Scene 3 (Seq3). (**d**) Scene 4 (Seq4). (**e**) Scene 5 (Seq5). (**f**) Scene 6 (Seq6).

Each image sequence consists of a series of images for each scene, Figure 6. Table 1 shows the specific information of Figure 6. The proposed method is compared with seven representative methods containing Tophat transform, LCM, MPCM, IPI, TV-PCP, PSTNN and SRWS.

**Table 1.** The specific information of the six image sequences.

| Sequence | Size | Number | Target Description | Background Description |
|---|---|---|---|---|
| 1 | 320 × 240 | 50 | Irregular shape<br>Low contrast | Cloudy background<br>Background changes quickly |
| 2 | 319 × 192 | 67 | Move quickly | Heavy noise<br>Bright background |
| 3 | 407 × 272 | 185 | Small<br>Vague and unclear | Complex background with trees |
| 4 | 298 × 186 | 40 | Tiny<br>Very low contrast | Dim background<br>Heavy noise |
| 5 | 320 × 240 | 200 | Small and bright<br>Slow-motion | Sea background with bridge |
| 6 | 332 × 221 | 300 | The cloud obscures the target<br>Size variation | Heavy cloud background<br>Clouds change quickly |

*4.3. Qualitative Results*

In this section, the experiments on six groups of sequential images are carried out to test the detection performance of the eight algorithms. Figures 8 and 9 show the detected target images of all detection methods of six image sequences. In particular, we mark targets in different scenes with red boxes. The three-dimensional views of gray images of target images are shown in Figures 10 and 11.

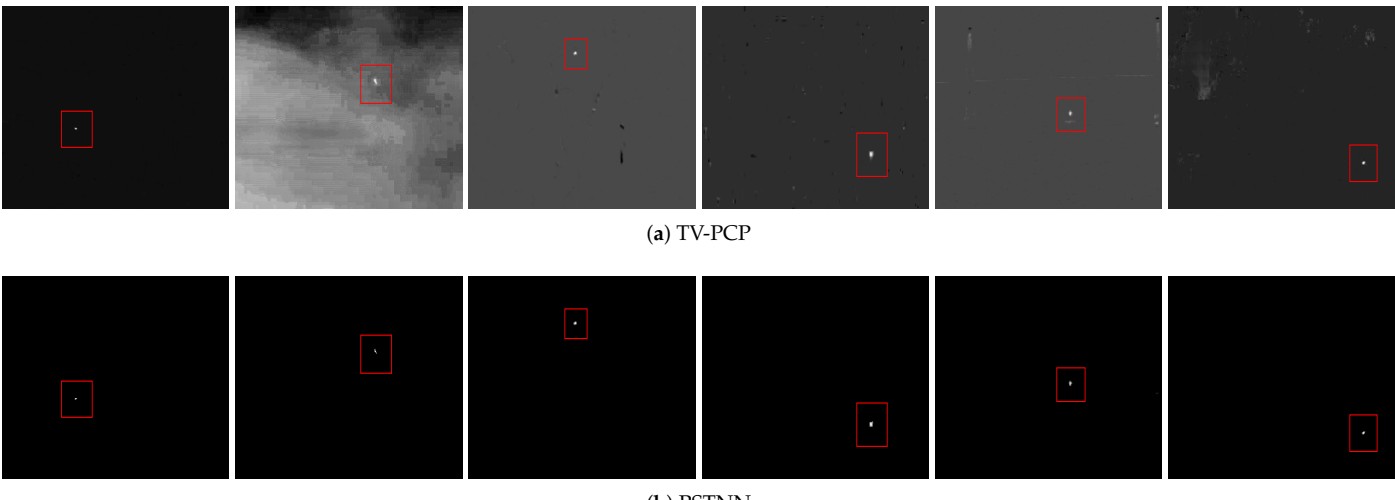

(**a**) Tophat

(**b**) LCM

(**c**) MPCM

(**d**) IPI

**Figure 8.** From left to right are the detection results of 1∼6 sequence images.

(**a**) TV-PCP

(**b**) PSTNN

**Figure 9.** *Cont.*

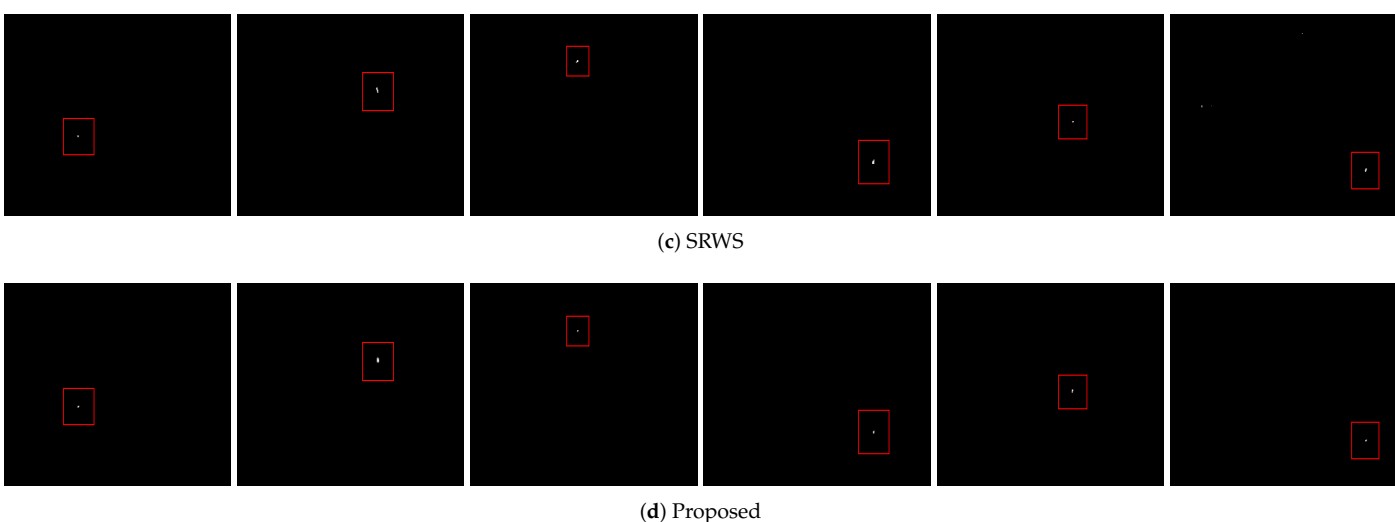

(**c**) SRWS

(**d**) Proposed

**Figure 9.** From left to right are the detection results of 1∼6 sequence images (Continued).

(**a**) Tophat

(**b**) LCM

(**c**) MPCM

(**d**) IPI

**Figure 10.** From left to right are the 3D gray images of the detection results of the 1∼6 sequences of images.

As shown in Figures 8 and 9, Tophat is particularly sensitive to noise and background edges. Under complex background clutters such as seq 2, seq 5 and seq 6, a large amount of

background information will be left in the target image, which results in a large false alarm rate. LCM can achieve fast and efficient detection. However, the targets are easily overwhelmed when the gray value of the target and background have trivial difference. The detection effect of IPI is good, but in the case of complex background and low signal-to-noise ratio, the target will not be detected and the accuracy of target detection cannot be guaranteed, as shown in seq 2 and seq 3. TV-PCP can recover the target well. As shown in seq 2, seq 5 and seq 6, due to the constraint of *TV* regularization, TV-PCP will produce the staircase effect, especially when the target moves rapidly. PSTNN and SRWS have good detection results, but when there is extremely rich background information, they will leave some non-target sparse points in the target image, which are difficult to distinguish from the real target, such as seq 5 and seq 6. Compared with the above seven baseline methods, our method can more effectively separate the background and target, has better target detection performance and can accurately estimate the background image and precisely detect the target.

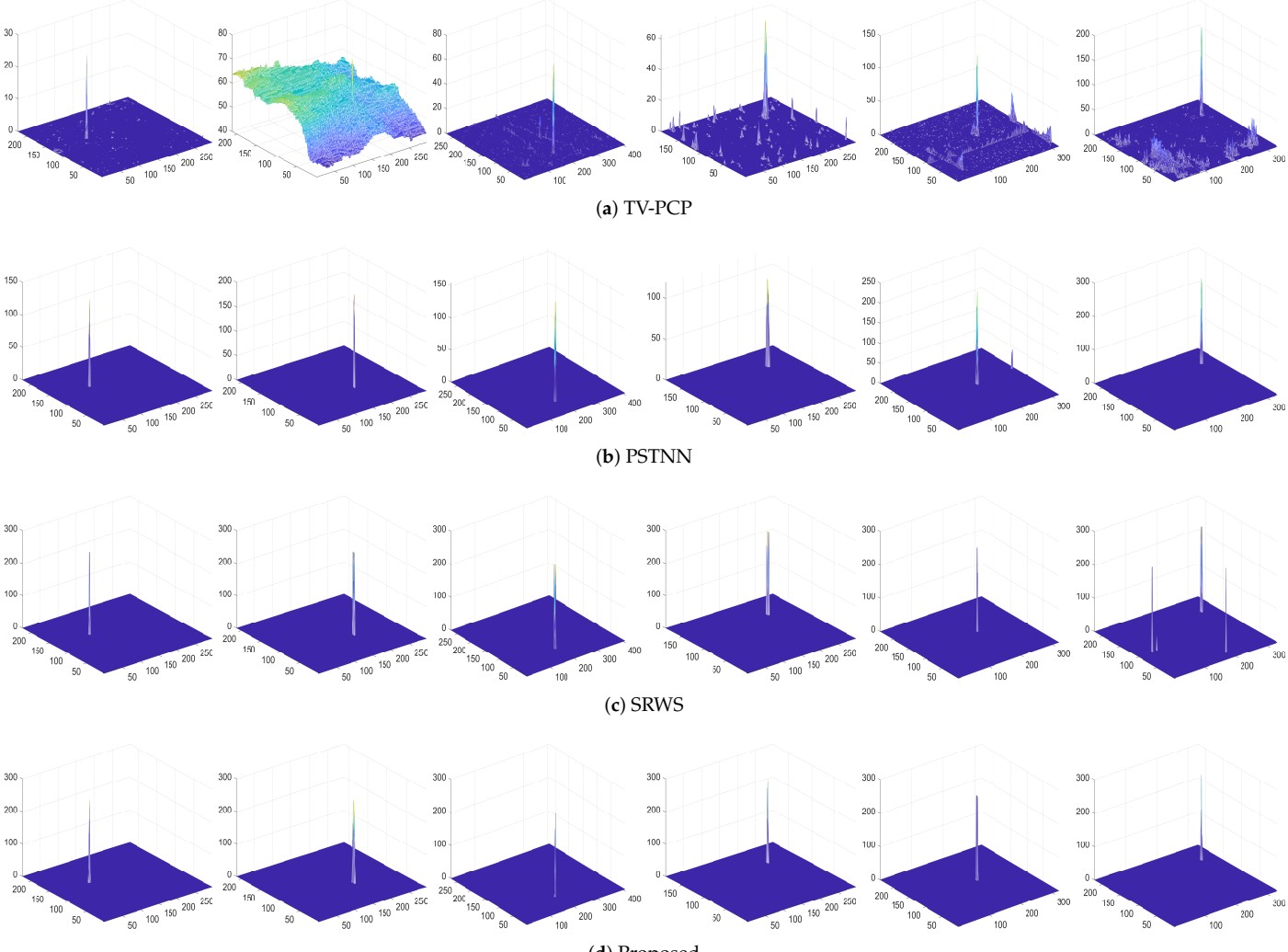

**Figure 11.** From left to right are the 3D gray images of the detection results of the 1~6 sequences of images (Continued).

As shown in Figures 10 and 11, in the case of a complex background, the detection results of Tophat, LCM, IPI and TV-PCP all have much background noise. The detection images of MPCM, PSTNN and SRWS are good, but in the particularly complex background, such as seq 5 and seq 6, the target image has multiple peaks and the detection is not accurate enough. As shown in Figures 8–11, our method has a better detection effect on various backgrounds, improves the accuracy of target detection and enhances the robustness of detection.

### 4.4. Quantitative Results

We quantitatively evaluate the detection effects of the eight algorithms in this part. SCRG and BSF are two important evaluation metrics. The specific results of all methods are shown in Table 2.

**Table 2.** SCRG and BSF of eight methods (bold red number: maximum value; bold blue number: second-highest value).

| Method | Seq1 SCRG | Seq1 BSF | Seq2 SCRG | Seq2 BSF | Seq3 SCRG | Seq3 BSF | Seq4 SCRG | Seq4 BSF | Seq5 SCRG | Seq5 BSF | Seq6 SCRG | Seq6 BSF |
|---|---|---|---|---|---|---|---|---|---|---|---|---|
| Tophat | 4.4421 | 4.6007 | 3.8941 | 3.8419 | 3.2609 | 3.3832 | 1.0877 | 1.1718 | 2.7220 | 2.7965 | 2.2323 | 2.2788 |
| LCM | 1.4192 | 0.6568 | 1.6625 | 0.7981 | 1.6013 | 0.5912 | 0.8250 | 0.2148 | 1.6755 | 0.4726 | 1.2586 | 0.2192 |
| MPCM | 7.2178 | 2.6165 | **7.3079** | 1.6609 | 2.6907 | 0.8652 | 0.9695 | 0.3156 | 1.9363 | 1.1390 | 1.4127 | 1.0858 |
| IPI | 7.2045 | 0.8598 | – | – | 3.8504 | 1.5899 | 1.4695 | 0.5808 | 4.2557 | 3.4883 | 2.6592 | 2.3908 |
| TV-PCP | 7.1253 | 8.0967 | 1.6206 | 1.5801 | 3.5875 | **4.3371** | 1.4335 | 1.8847 | 3.8874 | **4.8042** | 2.5588 | 2.8488 |
| PSTNN | 7.0204 | 1.5193 | **7.2397** | 0.7621 | 3.4487 | 1.8467 | 1.3657 | 1.0496 | 4.0430 | 2.6367 | 2.6204 | 2.0141 |
| SRWS | **19.5110** | **14.7651** | 5.6929 | **4.9813** | **4.2770** | 3.3859 | **5.6838** | **5.4759** | **4.7064** | 4.2971 | **5.0728** | **4.7346** |
| Proposed | **20.9825** | **15.8786** | 5.3570 | **4.6874** | **6.8538** | **5.4259** | **9.4167** | **9.0722** | **6.8972** | **6.2975** | **8.8536** | **8.2634** |

As shown in Table 2, due to the extremely low signal-to-noise ratio of the images in seq 2, IPI fails to detect and the corresponding metrics cannot be obtained. The evaluation metrics obtained by the proposed method are almost always the maximum or sub-maximum values. The SCRG and BSF are significantly improved compared with the other seven algorithms, which indicates that the proposed method can separate background and target well and has better background suppression and robustness.

A large number of experiments to test the ROC curve of each sequence image are made to more comprehensively evaluate the detection ability of each method and reflect the advantages of our method, as shown in Figure 12. Meanwhile, Table 3 summarizes the AUC of the ROC curve.

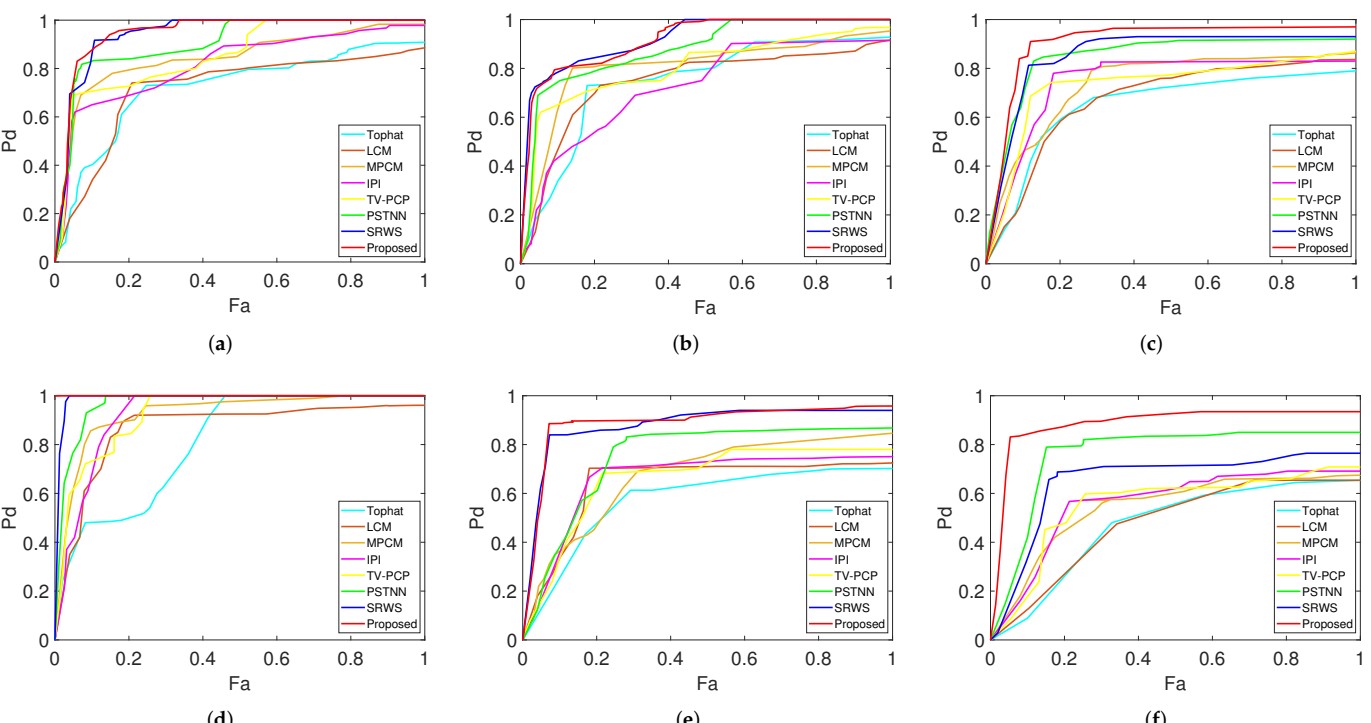

**Figure 12.** ROC curve of eight methods. (**a**) Seq1. (**b**) Seq2. (**c**) Seq3. (**d**) Seq4. (**e**) Seq5. (**f**) Seq6.

**Table 3.** AUC values of eight methods (bold red number: maximum value; bold blue number: second-highest value).

| Method | Seq1 | Seq2 | Seq3 | Seq4 | Seq5 | Seq6 |
|--------|------|------|------|------|------|------|
| Tophat | 0.7207 | 0.7473 | 0.6483 | 0.8082 | 0.5718 | 0.4698 |
| LCM | 0.7158 | 0.7432 | 0.6717 | 0.8661 | 0.6388 | 0.4761 |
| MPCM | 0.8458 | 0.8048 | 0.7372 | 0.9278 | 0.6693 | 0.5418 |
| IPI | 0.8174 | 0.7252 | 0.7442 | 0.9251 | 0.6579 | 0.5601 |
| TV-PCP | 0.8567 | 0.8032 | 0.7312 | 0.9247 | 0.6601 | 0.5531 |
| PSTNN | 0.9041 | 0.8784 | 0.8470 | 0.9677 | 0.7463 | **0.7604** |
| SRWS | **0.9428** | **0.9209** | **0.8571** | **0.9898** | **0.8780** | 0.6521 |
| Proposed | **0.9484** | **0.9168** | **0.9078** | **1.0000** | **0.8867** | **0.8850** |

It can be seen that Tophat and LCM have the worst performance; they cannot handle complex background images well. The target is easily submerged when MPCM processes cluttered and noisy images occurred. The detection performance of IPI is unstable, especially in seq 2; the accuracy of target detection is extremely poor. In the case of extraordinarily rich background information, TV-PCP cannot accurately predict the background image. PSTNN and SRWS have good detection performance, but the robustness of the algorithms cannot be guaranteed in the complex background of seq 5 and seq 6. Based on the above evaluation metrics, our method outperforms other methods in terms of background-suppression ability and target-detection ability and the robustness of our method to various complex backgrounds is proved.

We also calculate the running time for all methods in six sequence scenes. These experiments are all implemented on a computer with 16G of memory and an Intel Celeron 2.90 GHz CPU. As shown in Table 4, Tophat requires the least computation time and LCM and MPCM can also achieve fast detection because they all filter in the spatial domain. Both IPI and TV-PCP greatly increase the computational complexity, which requires a long computation time. With the premise of evaluating the background more accurately, we propose a solution strategy combined with ADMM, which simplifies the solution process, improves the convergence speed and greatly reduces the running time.

**Table 4.** Computation time (seconds) for all methods (bold red number: minimum value; bold blue number: second-smallest value).

| Method | Seq1 | Seq2 | Seq3 | Seq4 | Seq5 | Seq6 |
|--------|------|------|------|------|------|------|
| Tophat | **0.0012** | **0.0014** | **0.0011** | **0.0009** | **0.0010** | **0.0009** |
| LCM | **0.0964** | **0.0402** | **0.0443** | **0.0399** | **0.0304** | **0.0295** |
| MPCM | 0.2184 | 0.4967 | 0.4971 | 0.5490 | 0.5148 | 0.9059 |
| IPI | 39.6778 | 9.3576 | 55.9365 | 11.9990 | 30.4720 | 30.1724 |
| TV-PCP | 59.1727 | 15.4766 | 123.7174 | 64.4702 | 90.5041 | 88.1987 |
| PSTNN | 0.0974 | 0.0430 | 0.0539 | 0.0298 | 0.1032 | 0.0504 |
| SRWS | 0.9783 | 0.8261 | 2.4877 | 0.6208 | 1.3301 | 1.1024 |
| Proposed | 7.8637 | 7.3609 | 15.4934 | 6.2538 | 11.6141 | 8.6011 |

## 5. Discussion

Traditional filtering methods have a simple idea and a small amount of computation and only to some extent play a role in suppressing uniform background and cannot solve the problem of complex background. The methods based on human vision are mainly suitable for scenes where the target brightness is significantly different from the surrounding background. The optimization-based methods are obviously applicable to almost all kinds of complex and rapidly changing backgrounds and have strong robustness. The background data in the early IPI model is represented by the nuclear norm, which has good applicability to the background with slow change and uniformity, but still cannot deal with the image with complex background. The TV-PCP model improves the clarity of edges and corners and reduces noise interference, but there are residual dim edges in the

background image, showing a distinct staircase effect. PSTNN and SRWS fully consider the target characteristics, but their parameter settings limit the robustness. To improve the detection ability in complex backgrounds, we propose a TVWLR model.

The model introduces the TV regularization term to constrain the target to address the defect of the $l_1$ norm sparsity measurement and uses OEI to weight the background data to eliminate the obvious staircase effect. The proposed method is superior to other methods; Figures 8–11 show that our method has higher detection accuracy, Tables 2 and 3 demonstrate the robustness of our method to various complex backgrounds.

Although the proposed method has an excellent performance concerning detection ability, like other optimization-based algorithms, our method requires a lot of iterative operations. Compared with the traditional spatial domain algorithms, our method increases computational complexity and requires a slightly longer running time. Our future research work will focus on solving this problem. In addition, driven by big data and artificial intelligence, small target detection algorithms based on deep learning have made great progress. It is also a good idea to use the semantic segmentation model to detect small targets in complex backgrounds, which will also be our future research. The above experimental results indicate that, in the spatial domain algorithm, Tophat has extremely poor detection probability for complex background images. A lot of background clutter remains in the target image with LCM and missed detection and false detection occur in the process of MPCM. Compared with the spatial domain algorithm, the optimized detection algorithm has better detection performance. The IPI model has good detection performance in the case of uniform background, but it still cannot handle images with complex backgrounds. There are residual dim edges in the background image of the TV-PCP model, showing a distinct staircase effect. PSTNN and SRWS fully consider the target characteristics, but their parameter settings limit the robustness. The proposed method is superior to other methods, Figures 8–11 show that our method has higher detection accuracy, Tables 2 and 3 demonstrate the robustness of our method to various complex backgrounds.

Although the proposed method has an excellent performance concerning detection ability, like other optimization-based algorithms, our method uses iterative calculation. Compared with the traditional spatial domain algorithms, our method increases computational complexity and requires a slightly longer running time. Our future research work will focus on solving this problem.

## 6. Conclusions

In this paper, a new detection algorithm TVWLR is proposed to improve the detection accuracy of infrared dim small targets. The algorithm utilizes OEI to characterize the structural features of the image background and has the ability to adaptively determine the weight of the constraint low-rank term. It can suppress the staircase effect caused by the *TV* regularization term, enhance the details and edge information of the image and effectively reduce the false detection rate. The $l_{2,1}$ norm is introduced to remove strong edges and residuals in the image, which greatly improves the background suppression ability of the algorithm. Finally, we propose a solution algorithm combining ADMM for the TVWLR model. A large number of extensive experimental results demonstrate that the proposed method has better detection accuracy, better subjective and objective consistency and stronger robustness compared with the other seven methods.

**Author Contributions:** Conceptualization, X.C. and W.X.; Methodology, X.C.; Software, X.C. and Y.P.; Investigation, X.C. and T.G.; Formal Analysis, X.C.; Writing—Original Draft, X.C. and S.T.; Funding Acquisition, W.X.; Resources, W.X. and Q.F.; Supervision, W.X. and Q.F.; Writing—Review and Editing, W.X.; Data Curation, S.T.; Visualization, T.G.; Validation, Y.P. All authors have read and agreed to the published version of the manuscript.

**Funding:** This research was funded in part by the National Natural Science Foundation of China under Grant 62075219, 61805244, in part by the Key Technological Research Projects of Jilin Province, China under Grant 20190303094SF.

**Data Availability Statement:** The image data used in this paper are available at the following link: https://github.com/Tianfang-Zhang/SRWS and https://github.com/YimianDai/sirst, accessed on 25 December 2021.

**Conflicts of Interest:** The authors declare no conflict of interest.

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
