# Peer review of "Total Variation Weighted Low-Rank Constraint for Infrared Dim Small Target Detection"

_remotesensing, doi:10.3390/rs14184615_

Round 1
Reviewer 1 Report
Dear authors, I believe the paper is interesting but several issues should be addressed. My suggestions are as follows:
In terms of paper structure, I suggest the authors adopt more conventional section names, as suggested by this journal: Introduction, Materials and Methods, Results, Discussion, and Conclusion.
In the introduction, the authors mention several techniques for detecting dim small target detection but do not mention deep learning techniques (currently considered state-of-the-art in object detection). Several articles cover the use of deep learning:
Du, J., Lu, H., Hu, M., Zhang, L., & Shen, X. (2021). CNN‐based infrared dim small target detection algorithm using target‐oriented shallow‐deep features and effective small anchor. IET Image Processing, 15(1), 1-15.
Gao, Z., Dai, J., & Xie, C. (2019). Dim and small target detection based on feature mapping neural networks. Journal of Visual Communication and Image Representation, 62, 206-216.
Shi, M., & Wang, H. (2020). Infrared dim and small target detection based on denoising autoencoder network. Mobile Networks and Applications, 25(4), 1469-1483.
Wang, X., Peng, Z., Kong, D., & He, Y. (2017). Infrared dim and small target detection based on stable multisubspace learning in heterogeneous scene. IEEE Transactions on Geoscience and Remote Sensing, 55(10), 5481-5493.
Xue, D., Sun, J., Hu, Y., Zheng, Y., Zhu, Y., & Zhang, Y. (2020). Dim small target detection based on convolutinal neural network in star image. Multimedia Tools and Applications, 79(7), 4681-4698.
Zhou, S., Gao, Z., & Xie, C. (2022). Dim and small target detection based on their living environment. Digital Signal Processing, 120, 103271.
The data description is not clear to the readers. What do the authors mean by sequential images? How many images compose the sequences? Consider rewriting this part since it is confusing.
The discussion is too shallow. This section is very good for showing the readers a comparison with other studies and the relevance of what was made. The authors should thoroughly analyze and explain how this article is relevant. Besides, I think it would be a good idea to mention how deep learning could be used in this case or why would not it be a good idea. For example, it seems that a semantic/instance segmentation model could identify those small targets. This could be discussed or included as further experiments.
The authors could write possible future studies.
(Figure 5) It would be a good idea to put the coordinates of the image in examples a and b for easier comparison with the three-dimensional representations.
(Line 22) Since the authors mention that it “has been extensively studied”, maybe it is a good idea to insert more citations here on this topic.
(Line 30) “technology becomes” I suggest “technology has become”
(Line 32 and 40) This heading style is not correct. Either make it as text or subheadings (e.g., 1.1 and 1.2). Another good option would be to use all of that information and make a topic named Related works, which is widely adopted by many journals.
(Line 62) “target the and” I suggest “target and”
(Line 65) “the infrared patch image model (IPI) is created by Gao et al. [15],” I suggest “Gao et al. [15] created the infrared patch image model (IPI)”
(Line 69) “Zhang et al. introduced” I suggest keeping the style (line 65) “Zhang et al. [18], introduced”
(Line 72) Missing citation for “Wang et al. designed” “Wang et al. [19] designed”
(Line 74) Zhang et al. [20] used
(Line 88) “is able to better evaluate” I suggest “can better evaluate”
(Line 95) “given to accurately resolving the TVWLR model.” I suggest “given to resolve the TVWLR model accurately”
(Line 95) “And the solution process” I suggest “Moreover, the solution process”
(Line 137) Instead of writing “Ref [21]”, it sounds better to write like “Rudin et al [21]”
(Line 394) incomplete reference
(Line 405) incomplete reference
(Line 412) incomplete reference
(Line 422) “PP(99),” put in journal style
(Line 423) Incorrectly typed sentence “The title of the cited article”
(Line 432) incomplete reference
(Line 449) incomplete reference
Reviewer 2 Report
The authors proposed a method to improve detection of infrared small targets. The method is an extension to the existing infrared image patch model with the staircase effect removed. The authors have conducted experiments to evaluate the accuracy of the method and compared the result with eight other methods. The authors should address the following queries:
1. The authors should describe the acronyms before using them. There are a few places where the acronyms are either not defined or are used before explaining the full form of it.
2. The authors should proofread the article there are many typographical errors.
Reviewer 3 Report
The manuscript titled “Total variation weighted low-rank constraint for infrared dim small target detection” (Manuscript ID: remotesensing-1873136) presents a method, which is based on Total Variation Weighted Low Rank Constraint (TVWLR), for accurate infrared dim small target detection. My main concerns about the manuscript are:
1. The parameter setting part of the study should be expanded by giving references for the selected values? Is it possible to determine these values automatically?
2. The “Evaluation Metrics” were explained in Section 4 “4. Experiments and Analyses”. I think this part should be in Section 3.
3. What the SPIE (Page 1 Line 25) stands for? Please give initially long name and then initials. Please check the given reference (2) (Page 1 Line 25). Because the refered reference gave another references for this sentence.
4. Please give in the first time the long name of the “overlapping edge information” and then initials “(OEI)” (Page 3 Line 90). Then use the initials “OEI”
5. Please initially refer Table 4 in the text and then put the Table.
6. Please explain blue and red numbers In table 2 and 3 captions instead of text.
7. Please correct the "figure 2 caption as: “Figure 2. The IPI model detection results. (a) and (b) original images, (c) and (d) target images with clutter.”
8. In figure 6 and 7 captions, please give details about sub-figures (a, b, c, d, e and f)
9. Please give relation between the table 1 rows and Figure 6 sub figures.
Round 2
Reviewer 1 Report
The authors made the suggested changes.